# Prediction of 3-Year Survival in Patients with Cognitive Impairment Based on Demographics, Neuropsychological Data, and Comorbidities: A Prospective Cohort Study

**DOI:** 10.3390/brainsci13081220

**Published:** 2023-08-19

**Authors:** Dianxia Xing, Lihua Chen, Wenbo Zhang, Qingjie Yi, Hong Huang, Jiani Wu, Weihua Yu, Yang Lü

**Affiliations:** 1Department of Geriatrics, The First Affiliated Hospital of Chongqing Medical University, Chongqing 400016, China; dianxiaxing1982@163.com (D.X.);; 2Department of Geriatrics, Chongqing University Three Gorges Hospital, Chongqing 404100, China; 3Department of Quality Control, Chongqing University Three Gorges Hospital, Chongqing 404100, China; 4Institute of Neuroscience, Chongqing Medical University, Chongqing 400016, China

**Keywords:** cognitive dysfunction, dementia, nomograms, prognosis, survival rate

## Abstract

Objectives: Based on readily available demographic data, neuropsychological assessment results, and comorbidity data, we aimed to develop and validate a 3-year survival prediction model for patients with cognitive impairment. Methods: In this prospective cohort study, 616 patients with cognitive impairment were included. Demographic information, data on comorbidities, and scores of the Mini-Mental State Examination (MMSE), Instrumental Activities of Daily Living (IADL) scale, and Neuropsychiatric Inventory Questionnaire were collected. Survival status was determined via telephone interviews and further verified in the official death register in the third year. A 7:3 ratio was used to divide patients into the training and validation sets. Variables with statistical significance (*p* < 0.05) in the single-factor analysis were incorporated into the binary logistic regression model. A nomogram was constructed according to multivariate analysis and validated. Results: The final cohort included 587 patients, of whom 525 (89.44%) survived and 62 (10.56%) died. Younger age, higher MMSE score, lower IADL score, absence of disinhibition, and Charlson comorbidity index score ≤ 1 were all associated with 3-year survival. These predictors yielded good discrimination with C-indices of 0.80 (0.73–0.87) and 0.85 (0.77–0.94) in the training and validation cohorts, respectively. According to the Hosmer–Lemeshow test results, neither cohort displayed any statistical significance, and calibration curves displayed a good match between predictions and results. Conclusions: Our study provided further insight into the factors contributing to the survival of patients with cognitive impairment. Clinical Implications: Our model showed good accuracy and discrimination ability, and it can be used at community hospitals or primary care facilities that lack sophisticated equipment.

## 1. Introduction

Global aging leads to various forms of functional decline, and cognitive impairment is one of the most disruptive changes. Cognitive impairment includes varying degrees of cognitive decline, ranging from mild cognitive impairment (MCI) to dementia. Since the disease is incurable and progressive, planning management strategies well in advance is becoming increasingly important. Clinical guidelines on cognitive impairment advise factoring the patient’s life expectancy into clinical decisions [1]. It is important to provide appropriate support according to patients’ life expectancy so that the needs and aspirations of patients are met. A prediction model is one of the most useful tools for estimating patients’ life expectancy. To date, some prediction models have been developed to determine survival in patients with cognitive decline based on cerebrospinal fluid [2], blood biological markers [3], magnetic resonance imaging (MRI) [4] or positron emission tomography–computed tomography (PET-CT) findings [5]. However, the practical application of these predictors is limited. For example, to obtain cerebrospinal fluid samples for the subsequent analysis of biomarkers, patients are required to undergo lumbar puncture. However, certain patients, particularly those exhibiting neuropsychiatric symptoms such as delusions and hallucinations, may be unable to fully cooperate in the collection of cerebrospinal fluid. Similarly, the use of MRI data for prediction purposes is better suited for the collaboration of proficient radiologists than for community screening scenarios. PET-CT is only available in advanced medical centers and is unaffordable for some patients, given that it may not always be covered by insurance, as is the case with national health insurance in China. Similarly, the detection of blood biomarkers requires sophisticated equipment that may not always be available in the primary care facilities visited by such patients. Furthermore, the current guidelines do not recommend biomarker testing as a routine procedure [1]. The practical application of these models is therefore limited.

The comprehensive neuropsychological assessment of patients with cognitive decline is essential to identifying cognition impairment and to planning subsequent treatment and care; these assessment data are available for almost every single patient with cognitive impairment. Scales such as the Mini-Mental State Examination (MMSE) [6], Instrumental Activities of Daily Living (IADL) Scale [7], and Neuropsychiatric Inventory (NPI) [8] reveal a wide range of disease characteristics. The MMSE is widely used as a cognitive screening instrument [9] owing to its simplicity and robust validity [6]. It can be employed in clinical environments and as a screening tool for cognitive impairment in community-dwelling populations. Prior research has indicated that cognitive limitations increase the risk of mortality in patients with hip fracture [10]. Clear, the direct correlation between MMSE score and the survival of patients with cognitive impairment remains uncertain. IADL is frequently used to evaluate functional performance. Substantial research has substantiated the association between reduced IADL performance and adverse outcomes, such as readmission and transition from MCI to dementia [11]. However, the relationship between the IADL performance and patient prognosis remains unclear. The NPI has been employed in the domain of neuropsychiatric symptoms pertaining to neurodegenerative disorders for approximately three decades. The survey has been translated into over 40 languages, and its credibility and dependability have been substantiated through numerous investigations [12]. The existing literature has examined the correlation between the NPI and prognosis [13]; however, certain gaps remain. These gaps include determining the specific symptoms linked to prognosis and investigating whether the combination of the NPI with other neuropsychological testing scales can enhance the accuracy of patient survival predictions.

## 2. Materials and Methods

### 2.1. Design and Setting

The study covered the period from January 2012 to December 2017 at our hospital’s memory clinic. Demographic data and information about comorbidities were extracted from medical records. All the patients underwent neuropsychological assessments performed by professional neuropsychological evaluators during their first visit. The diagnosis was performed jointly by neurologists and geriatricians according to the National Institute on Aging and Alzheimer’s Association guidelines and the Chinese Guidelines for the Diagnosis and Treatment of Dementia, which were both published in 2011 [14,15,16,17]. Since the estimated median survival time was 3.3 years for patients with dementia [18], we set the time point for acquiring the patients’ survival status through telephone interview at the third year after their first neuropsychological test. If a patient was reported to have died within 3 years, we further verified this information in the official death register.

### 2.2. Ethical Considerations

The Declaration of Helsinki was followed during the course of conducting this study. The Ethics Committee of The First Affiliated Hospital of Chongqing Medical University approved this study (approved on 22 July 2014; approval no. 2014-15-2). Participants or their families provided written informed consent for study participation.

### 2.3. Patients

The MMSE was used to screen patients exhibiting cognitive impairment. According to a previous report, MMSE scores were related to education level [19]. Consequently, we relied on established criteria from previous scholarly studies to identify individuals with cognitive impairment: MMSE score < 18 points in illiterate patients, MMSE score < 24 points in patients with 1–11 years of education, and MMSE scores 27–30 points in those with more than 11 years of education [9]. Patients who declined to provide data for the study, refused to be interviewed by telephone, or were unable to be reached 3 years after the first visit were excluded from the study.

### 2.4. Neuropsychological Assessment

Cognitive function was screened using the Chinese version of the MMSE. The validity and reliability of this scale in the Chinese population were confirmed in a previous study [20]. Better cognitive function is indicated by higher scores. The Lawton and Brody IADL scale was used to evaluate the activities of daily living. The scale comprises assessments of shopping, using public transportation, cooking, housekeeping, laundry, utilizing the telephone, administering medication, and managing finances. A higher IADL score indicates a lower ability to perform daily activities. The NPI questionnaire was used to evaluate neuropsychiatric symptoms (NPS). Each subitem was scored by multiplying its frequency (from 1 to 4, higher scores indicate greater frequency) with its severity (from 1 to 3, higher scores indicate more severe conditions); zero represented the absence of symptoms. All evaluation data were reviewed and validated by experienced neurologists and geriatricians. Cognitive impairment diagnoses were further confirmed by proficient physicians who employed clinical data and a thorough neuropsychological evaluation.

### 2.5. Charlson Comorbidity Index (CCI) Calculation

Following neuropsychological assessments, the comorbidity diagnoses of patients exhibiting cognitive impairment were obtained from the electronic medical record system in accordance with the International Classification of Diseases, Tenth Revision (ICD-10). Subsequently, the diagnosis was further validated by senior doctors at our hospital. After this validation process, the CCI score was calculated for each patient who presented with cognitive impairment. The CCI encompasses a set of 19 items that align with diverse comorbidities, with each item assigned a distinct weight to account for the adjusted risk of mortality within a year. The cumulative CCI score is derived by summing these weights, wherein higher scores signify diminished odds of survival and greater severity of comorbidity [21].

### 2.6. Data Preparation

Data on age, sex, education years, MMSE, IADL, and NPI subitem scores, and complications were collected. Predictors were selected based on previous research [22,23]. Among these variables, age, education years, and MMSE and IADL scores were continuous variables, and sex was a dichotomous variable. The 12 subitem scores of the NPI were converted into dichotomous scores based on whether or not each subitem had a score above 0. If the score was >0, it was registered as 1 point; if the score was 0, it was retained as such. Similarly, the CCI was converted into dichotomous values: if the CCI was >1, it was recorded as 1; if it was ≤1, it was recorded as 0.

### 2.7. Statistical Analyses

A 7:3 ratio was used to divide the cohort into the training and validation datasets. The Shapiro–Wilk test was used to test for normal distribution. First, single-factor analysis was used to analyze each variable according to the distribution. Two-sample *t* tests were used to analyze normally distributed variables, Wilcoxon rank sum tests were used to analyze non-normally distributed variables, and Chi-square tests were used for categorical variables. Variables with a significance level of *p* < 0.05 were incorporated into the multivariate binary logistic regression model. A nomogram was then constructed according to multivariate analysis. Furthermore, receiver operating characteristic (ROC) curves were constructed to assess the discriminative capability of the model in the training and validation sets. Subsequently, the area under the curve (AUC) and concordance index (C-index) were calculated, calibration curves were drawn, and the Hosmer–Lemeshow test was used to determine the model’s accuracy. R software was used for statistical analysis (version 4.0.3, https://www.r-project.org, accessed on 1 December 2020).

## 3. Results

### 3.1. Study Flowchart

The study flowchart is displayed in Figure 1.

### 3.2. Study Population

A total of 616 patients were initially enrolled, of which 29 were lost to follow-up in the third year. Therefore, the final number of the study cohort was 587. There were 349 women and 238 men with a median age of 76 years (interquartile range [IQR]: 68–82), median years of education of 9 (IQR: 6–12), median MMSE score of 18 (IQR: 10–25), and median IADL score of 14 (IQR: 10–21). By the end of the third year after their first visit, 525 patients were still alive (89.44%). Among all patients with cognitive decline, 310 patients were diagnosed with dementia due to Alzheimer’s disease (AD), 243 with MCI due to AD, 13 with vascular dementia, 11 with mixed dementia, and 10 with other types of dementia, including dementia with Lewy bodies, alcoholic dementia, and unclassifiable dementia.

### 3.3. Training of the Prediction Model

Of the 587 eligible patients, 411 and 176 were included in the training and validation sets, respectively; their data are summarized and compared in Table 1. No statistically significant differences were detected between the training and validation sets in any of the independent variables. Age, MMSE score, IADL score, disinhibition, and the CCI were significantly different between the deceased and surviving patients in the training set (Table 2). Factors with statistically significant differences were incorporated into the multivariate binary logistic regression model, and a nomogram was developed (Figure 2). The nomogram indicated that younger age, higher MMSE scores, lower IADL scores, absence of disinhibition, and CCI ≤ 1 were protective factors for the 3-year survival of patients with cognitive impairment.

### 3.4. Validation of the Prediction Model

The ROC curves demonstrated the good discriminatory ability of the prediction models (Figure 3) with AUCs of 0.80 (95% confidence interval: 0.73–0.87) and 0.85 (95% confidence interval: 0.77–0.94) in the training and validation sets, respectively. No statistical significance was found in the Hosmer–Lemeshow test results (*p* = 0.94 in the training cohort, *p* = 0.83 in the validation cohort). Taken together, the Hosmer–Lemeshow test results and the calibration plots indicated that both training and validation cohorts agreed well with predicted data (Figure 4).

## 4. Discussion

In this study, we developed and validated a prediction model for the 3-year survival of patients with cognitive impairment based on demographics, neuropsychological assessment data, and comorbidities. Our model revealed that younger age, higher MMSE scores, lower IADL scores, the absence of disinhibition, and CCI ≤ 1 were protective factors for the 3-year survival of patients with cognitive impairment. The good accuracy and discriminative ability of our model confirmed the feasibility of combining neuropsychological test data, clinical information, and demographic data to assess the 3-year survival in patients with cognitive impairment. Considering the accessibility of these predictors for most hospitals, our model can be utilized in primary health care facilities or hospitals that lack equipment such as MRI and PET-CT. In the aforementioned contexts, our model employs a concise set of five variables, including age, MMSE score, IADL score, presence or absence of disinhibition symptoms, and CCI score > 1 or not, to estimate a patient’s chances of surviving for three years. This estimation can aid in identifying patients with lower odds of survival, thereby facilitating treatment decisions, care provision, and the development of social support plans.

Some predictive models have been established previously. These models were applied to patients that differed in their characteristics. The Advanced Dementia Prognostic Tool, for example, was applied for the prediction of mortality in patients with advanced dementia residing in nursing homes [24]. Another model exhibited an exceptional mortality prediction performance for dementia [25]. However, it still had some limitations. This model indiscriminately incorporated data from all the health records into machine learning, some of which were irrelevant to the research subject. By contrast, our model was developed based on demographic information and neuropsychological test data; all independent variables were readily available in general practice and interpreted. Moreover, the external validation of our model in other patient cohorts can easily be performed.

In our study, age, MMSE score, and IADL score primarily influenced the 3-year survival of patients with cognitive impairment. The finding that younger age was a protective factor was consistent with the findings of a previous study [26]. The MMSE is currently the most widely used scale for cognitive assessment. Previous studies on the MMSE have shown that both sensitivity and specificity of the MMSE in distinguishing healthy elderly people from those with dementia exceed 80% [27,28]. However, the use of MMSE in the MCI population is still limited. The reason for this is that the MMSE scale is not sensitive to the decline in advanced living ability, while patients with MCI are characterized by normal basic cognitive function and declining advanced living ability. Therefore, the use of MMSE remains limited in those with MCI. Thus, the IADL scale included in our study can compensate for this deficiency. The IADL scale includes assessments of the skills necessary to maintain self-care, health and social support, and activities that help create social identity. Previous research indicated that poorer IADL performance is linked to faster and higher conversion rates into dementia and stronger predictions of future dementia [29]. Our study contributes to the evidence that a worse IADL score is associated with fewer opportunities for 3-year survival in patients with cognitive impairment.

A previous study found that total NPI scores are associated with mortality in those with cognitive impairment [30]. However, if the neuropsychological symptoms were analyzed through the total NPI score for their effect on patient outcomes, the results indicated that each subitem had an equal role and incidence rate. By analyzing every NPI subitem, we found that only disinhibition accounted for a significant influence on survival. Previous studies of NPS in patients with cognitive decline mainly focused on delusion, agitation, and other behavioral and psychological symptoms [31]; studies on disinhibition are rare. This finding of our model contributes to evidence for the prognostic significance of disinhibition in patients with cognitive impairment. Therefore, intervention for disinhibition could be an important part of treatment. Furthermore, disinhibition can amplify the probability of caregiver fatigue or ailment, consequently intensifying susceptibility to patient mistreatment, institutionalization, and the subsequent escalation of caregiver depression [32,33,34]. This finding implies the necessity of enhancing risk management for patients exhibiting disinhibition symptoms and providing suitable caregiver support while formulating long-term care strategies. Interestingly, apathy, delusion, and hallucination were previously shown to adversely affect the prognosis of patients with cognitive impairment [35]. These symptoms were correlated with more serious living ability impairment, accelerated cognitive decline, earlier institutionalization, and greater frequency of hospitalization [36]. However, in our cohort, no differences were detected between surviving and deceased patients regarding these symptoms. A possible reason is that we did not screen the disease stage of AD patients, who accounted for 52.8 percent of our cohort; however, the stage of AD can have a significant impact on the incidence of NPS and patients’ survival [37,38]. From this point of view, it can also be said that our model is more suitable for use in primary health care facilities in cases where the stage of cognitive impairment remains unclear.

This study supports the findings of previous study, i.e., that the CCI is a prognostic factor [39]. However, there are still some differences between our data processing and previous studies. The neurologists and geriatricians in our team supposed that patients with a score of 1 could be treated well in most cases and may have a better survival rate than patients with poor CCI scores. Furthermore, the CCI was developed based on 1-year mortality [40]; due to the longer follow-up period in this study, patients with higher CCI scores were less likely to survive. Therefore, we dichotomized the CCI score with a 1-point threshold. As expected, the nomogram revealed that CCI ≤ 1 was a protective factor. New evidence was provided by our study for the prognostic significance of CCI, as well as evidence that treating comorbidities of patients with cognitive impairment may help to decrease economic burden, care burden, and medical burden [41].

This study has some limitations. First, owing to the significance of neuropsychological testing in our model, patients who were unable to engage in such testing, either due to concurrent hearing impairment or other factors, were precluded from utilizing our model to predict 3-year survival. Second, the nomogram was based on data obtained exclusively from our memory clinic, which introduced a potential selection bias. Conducting external validation in a heterogeneous patient population encompassing individuals with diverse cultural and social backgrounds, varying comorbidities, and those seeking care at alternative memory clinics is of the utmost importance for enhancing the generalizability and robustness of our model. Third, it is important to acknowledge that the scales employed in our research, such as the MMSE, have been developed into several versions across different regions. Consequently, it would be inappropriate to apply our predictive model to patients who were assessed using different versions of the scale. Fourth, we were unable to obtain the exact date of death of most deceased patients at the 3-year follow-up because of cultural norms. We performed binary logistic regression instead of Cox regression. Our model could predict the survival status of patients exclusively at the third-year time point, whereas it was unable to accurately predict the survival time for individuals with cognitive impairment. Fifth, the target of our study was to establish a model for primary care facilities or hospitals that lack advanced equipment. The clinical stage of patients who visited those health facilities remained unspecified; however, different stages may lead to divergences in the survival time of patients with cognitive impairment [42]. There is a possibility of bias when our model is applied to patients with varying ratios of cognitive impairment. Consequently, it is imperative to conduct additional verifications of the generalizability of our model across different patient cohorts. Finally, it is important to acknowledge the effects of treatment and care decisions on patient survival. However, in the interests of efficiency and simplicity, we decided to exclude these factors from our model because of their inherent variability [1]. The future research directions encompass the following aspects: augmenting the size of the training dataset to enhance the resilience of the model; conducting external validation across diverse patient populations with cognitive impairment to bolster the model’s generalizability; incorporating factors such as treatment and nursing that impact patient survival to enhance the model’s efficacy; extending the follow-up period to ascertain the precise survival duration of a larger cohort of patients; and subsequently establishing a Cox regression survival prediction model.

## 5. Conclusions

In conclusion, we provide further evidence that disinhibition is associated with poor prognoses in patients with cognitive impairment. We developed and validated a nomogram based on readily available data, and our model achieved excellent accuracy and consistency for internal validation.

## Figures and Tables

**Figure 1 brainsci-13-01220-f001:**
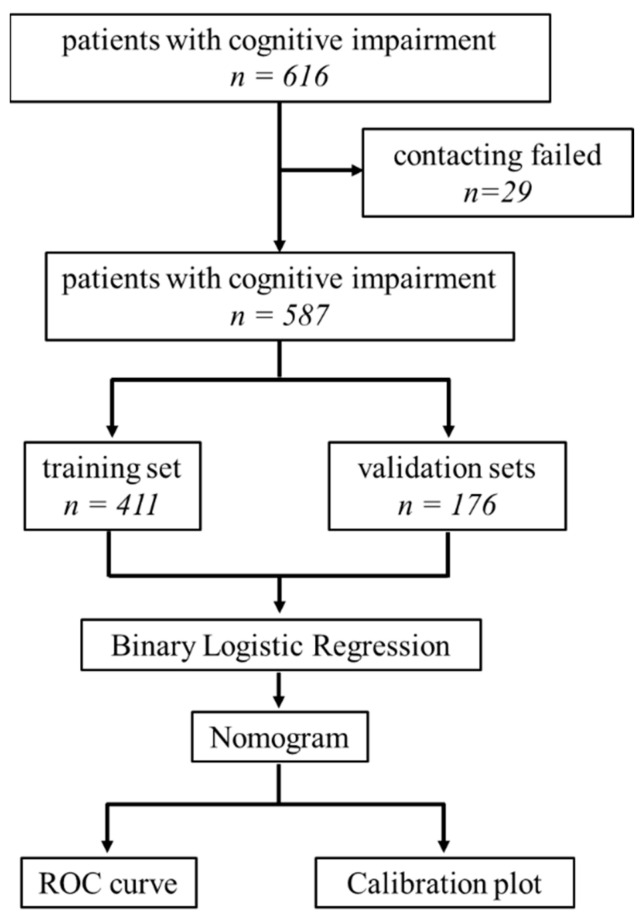
Study flowchart. ROC curve: receiver operator characteristic curve.

**Figure 2 brainsci-13-01220-f002:**
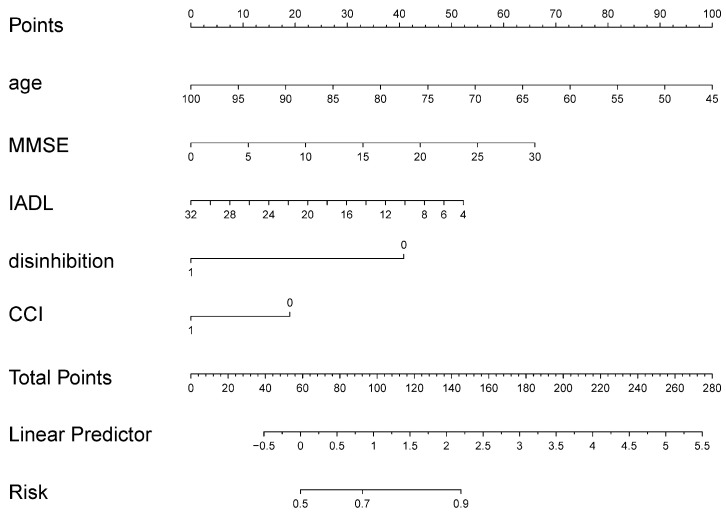
The developed nomogram for predicting 3-year survival in the training cohort. Each covariate was represented by a vertical line, and the corresponding score at the intersection of the vertical and points line was recorded and observed. This process was iterated until a point value was obtained for each covariate, and the cumulative sum of all points yielded the total number of points. Subsequently, a vertical line was drawn downwards from the total point line, and the value at the intersection of the risk line at the bottom indicated the patient’s probability of survival for three years. Disinhibition: 0, absence of disinhibition; 1, presence of disinhibition. CCI: 0, ≤1; 1, >1.

**Figure 3 brainsci-13-01220-f003:**
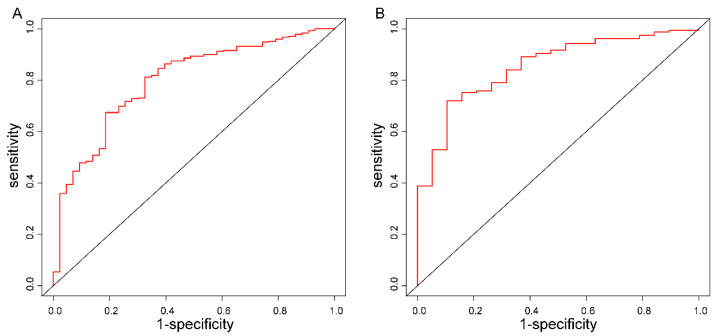
ROC of the model in the training (**A**) and validation (**B**) cohorts. The AUCs are 0.797 (0.728–0.866) and 0.853 (0.772–0.935) in the training and validation cohorts, respectively.

**Figure 4 brainsci-13-01220-f004:**
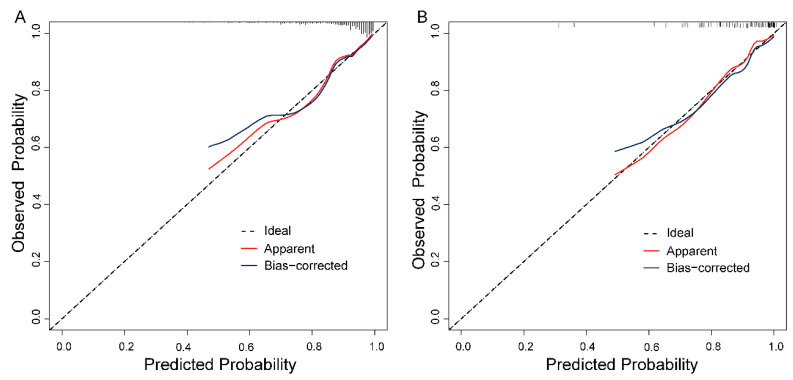
Calibration plot of the model in the training (**A**) and validation (**B**) cohorts. The *x* and *y* axes represent the predicted probability of 3-year survival and the actual result, respectively. The diagonal line represents the ideal prediction. The apparent line (red) represents the predicted probabilities corresponding to the observed; the bias-corrected line (blue) represents the model’s predictive capacity after bootstrap correction.

**Table 1 brainsci-13-01220-t001:** Comparisons between the training and validation sets.

	Level	Overall	Training	Validation	*p*
*n*		587	411	176	
Sex (%)	male	238 (40.5%)	168 (40.9%)	70 (39.8%)	0.88
	female	349 (59.5%)	243 (59.1%)	106 (60.2%)	
Survival (%)	deceased	62 (10.6%)	43 (10.5%)	19 (10.8%)	1.00
	living	525 (89.4%)	368 (89.5%)	157 (89.2%)	
Delusion (%)	no	446 (76.0%)	313 (76.2%)	133 (75.6%)	0.96
	yes	141 (24.0%)	98 (23.8%)	43 (24.4%)	
Hallucination (%)	no	458 (78.0%)	318 (77.4%)	140 (79.5%)	0.64
	yes	129 (22.0%)	93 (22.6%)	36 (20.5%)	
Agitation (%)	no	491 (83.6%)	337 (82.0%)	154 (87.5%)	0.13
	yes	96 (16.4%)	74 (18.0%)	22 (12.5%)	
Depression (%)	no	442 (75.3%)	308 (74.9%)	134 (76.1%)	0.84
	yes	145 (24.7%)	103 (25.1%)	42 (23.9%)	
Anxiety (%)	no	433 (73.8%)	297 (72.3%)	136 (77.3%)	0.25
	yes	154 (26.2%)	114 (27.7%)	40 (22.7%)	
Euphoria (%)	no	568 (96.8%)	399 (97.1%)	169 (96.0%)	0.68
	yes	19 (3.2%)	12 (2.9%)	7 (4.0%)	
Apathy (%)	no	396 (67.5%)	275 (66.9%)	121 (68.8%)	0.73
	yes	191 (32.5%)	136 (33.1%)	55 (31.2%)	
Disinhibition (%)	no	529 (90.1%)	366 (89.1%)	163 (92.6%)	0.24
	yes	58 (9.9%)	45 (10.9%)	13 (7.4%)	
Irritability (%)	no	394 (67.1%)	276 (67.2%)	118 (67.0%)	1.00
	yes	193 (32.9%)	135 (32.8%)	58 (33.0%)	
Aberrant motor	no	510 (86.9%)	353 (85.9%)	157 (89.2%)	0.34
behavior (%)	yes	77 (13.1%)	58 (14.1%)	19 (10.8%)	
Night behavior (%)	0	458 (78.0%)	324 (78.8%)	134 (76.1%)	0.54
	1	129 (22.0%)	87 (21.2%)	42 (23.9%)	
Appetite/eating (%)	0	532 (90.6%)	367 (89.3%)	165 (93.8)	0.12
	1	55 (9.4%)	44 (10.7%)	11 (6.2%)	
CCI (%)	0	370 (63.0%)	264 (64.2%)	106 (60.2%)	0.41
	1	217 (37.0%)	147 (35.8%)	70 (39.8%)	
Age (median [IQR])		76.00 [68.00, 82.00]	76.00 [69.00, 82.00]	75.00 [67.00, 80.00]	0.13
MMSE (median [IQR])		18.00 [10.00, 25.00]	18.00 [9.00, 24.00]	19.00 [11.00, 25.00]	0.23
Education years (median [IQR])		9.00 [6.00, 12.00]	9.00 [6.00, 12.00]	9.00 [6.00, 12.00]	0.25
IADL (median [IQR])		14.00 [10.00, 21.00]	14.00 [10.00, 21.00]	14.00 [10.00, 21.00]	0.86

Data are expressed as *n* (%) or median [interquartile range]. *p* values were calculated using χ^2^ test for categorical data; Wilcoxon rank-sum test was used for continuous data. Abbreviations: CCI, Charlson Comorbidity Index; MMSE, Mini-Mental State Examination; IADL, Instrumental Activity of Daily Living; IQR, interquartile range.

**Table 2 brainsci-13-01220-t002:** Single factor comparative analysis between deceased and alive subjects in the training cohort.

	Level	Overall	Deceased	Living	*p*
*n*		411	43	368	
Sex (%)	male	168 (40.9%)	20 (46.5%)	148 (40.2%)	0.53
	female	243 (59.1%)	23 (53.5%)	220 (59.8%)	
Delusion (%)	no	315 (76.6%)	30 (69.8%)	285 (77.4%)	0.35
	yes	96 (23.4%)	13 (30.2%)	83 (22.6%)	
Hallucination (%)	no	330 (80.3%)	32 (74.4%)	298 (81.0%)	0.41
	yes	81 (19.7%)	11 (25.6%)	70 (19.0%)	
Agitation (%)	no	350 (85.2%)	34 (79.1%)	316 (85.9%)	0.34
	yes	61 (14.8%)	9 (20.9%)	52 (14.1%)	
Depression (%)	no	301 (73.2%)	33 (76.7%)	268 (72.8%)	0.71
	yes	110 (26.8%)	10 (23.3%)	100 (27.2%)	
Anxiety (%)	no	301 (73.2%)	28 (65.1%)	273 (74.2%)	0.28
	yes	110 (26.8%)	15 (34.9%)	95 (25.8%)	
Euphoria (%)	no	399 (97.1%)	39 (90.7%)	360 (97.8%)	0.032 *
	yes	12 (2.9%)	4 (9.3%)	8 (2.2%)	
Apathy (%)	no	276 (67.2%)	25 (58.1%)	251 (68.2%)	0.25
	yes	135 (32.8%)	18 (41.9%)	117 (31.8%)	
Disinhibition (%)	no	372 (90.5%)	30 (69.8%)	342 (92.9%)	<0.001 ***
	yes	39 (9.5%)	13 (30.2%)	26 (7.1%)	
Irritability (%)	no	272 (66.2%)	29 (67.4%)	243 (66.0%)	0.99
	yes	139 (33.8%)	14 (32.6%)	125 (34.0%)	
Aberrant motor behavior (%)	no	357 (86.9%)	38 (88.4%)	319 (86.7%)	0.94
	yes	54 (13.1%)	5 (11.6%)	49 (13.3%)	
Night behavior (%)	no	319 (77.6%)	31 (72.1%)	288 (78.3%)	0.47
	yes	92 (22.4%)	12 (27.9%)	80 (21.7%)	
Appetite/eating (%)	no	375 (91.2%)	41 (95.3%)	334 (90.8%)	0.47
	yes	36 (8.8%)	2 (4.7%)	34 (9.2%)	
CCI (%)	≤1	254 (61.8%)	20 (46.5%)	234 (63.6%)	0.044 *
	>1	157 (38.2%)	23 (53.5%)	134 (36.4%)	
Age (median [IQR])		75.00 [68.00, 80.50]	79.00 [75.50, 84.50]	75.00 [67.75, 80.00]	<0.001 ***
MMSE (median [IQR])		19.00 [10.00, 25.00]	9.00 [5.00, 15.00]	19.00 [11.75, 25.00]	<0.001 ***
Education years (median [IQR])		9.00 [6.00, 12.00]	9.00 [3.00, 9.50]	9.00 [6.00, 12.00]	0.12
IADL (median [IQR])		14.00 [10.00, 21.00]	21.00 [15.50, 27.50]	13.00 [9.75, 20.00]	<0.001 ***

*: *p* < 0.05; ***: *p* < 0.001. Abbreviations: CCI, Charlson Comorbidity Index; MMSE, Mini-Mental State Examination; IADL, Instrumental Activity of Daily Living; IQR, interquartile range.

## Data Availability

All data generated or analysed during this study are included in this article. Further enquiries can be directed to the corresponding author.

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
