# Peer review of "Prediction of 3-Year Survival in Patients with Cognitive Impairment Based on Demographics, Neuropsychological Data, and Comorbidities: A Prospective Cohort Study"

_brainsci, 2023, doi:10.3390/brainsci13081220_

Round 1

Reviewer 1 Report

The main question is whether researchers can predict how long memory disorder patients will live based on a variety of key variables. 

A well-designed, well-executed, and appropriately presented study.

Author Response

Point 1: The main question is whether researchers can predict how long memory disorder patients will live based on a variety of key variables. 

Response 1:

We express our gratitude for your inquiry. Our model was unable to predict the survival duration of patients with memory impairment. Reasons are as follows: Firstly, median survival time from age of onset of dementia ranges from 3.3 to 11.7 years, with most studies in the 7 to 10-year period [1], the duration of our study spanned a period of six years, commencing in 2012 and concluding in 2017. Consequently, we could only ascertain the survival status at the conclusion of the 3-year observation period instead of the precise survival durations. Secondly, as explicated in our research paper, diverse cultural and other variables impeded us to acquire accurate survival durations for all patients, particularly those who had deceased by the termination of the third year of observation. As a result, our subsequent assessment was constrained to a dichotomous occurrence indicating either survival or non-survival at the three-year juncture. Thus, the primary aim of our investigation was to predict the probability of survival three years post-diagnosis.

Point 2: A well-designed, well-executed, and appropriately presented study.

Response 2: Thank you for your positive comments.

Reviewer 2 Report

General comment:

In this article, the authors created a 3-year survival prediction model for patients with cognitive impairment using readily available demographic data, neuropsychological assessment results, and comorbidity information. For this, the authors included 616 patients in their study. Of which in the final cohort of 587 patients, 525 (89.44%) survived and 62 (10.56%) died. Demographic information, comorbidity data, and Mini-Mental State Examination (MMSE), Instrumental Activities of Daily Living (IADL) scale, and Neuropsychiatric Inventory Questionnaire scores were collected. Survival status was determined by telephone interviews and further verified in the official death registry in the third year.

Based on the results obtained, the authors indicated that their study provides valuable insights into the factors affecting the survival of patients with cognitive impairment. The predictive model they developed shows high accuracy and discrimination ability, making it suitable for use in community hospitals or primary care settings without advanced equipment.

In general the text is well written, clear and orderly. The topic is an interesting one and falls within the theme of the journal Brain Sciences. The manuscript is scientifically sound and the results can be reproduced.

The results of the manuscript are of average novelty, the authors previously made visible part of the results presented in the paper.

https://doi.org/10.1002/alz.062850

Figures/tables/images/diagrams are appropriate. But it is suggested to draw up an experimental design in the form of an image/Scheme, for a better understanding of the experimental procedures.

The conclusions are in agreement with the results of the study.

Minor corrections:

RESULTS

Alzheimer's disease (AD) - one line below is written Alzheimer's disease. It is suggested to use the abbreviation after its explanation.

Author Response

Point 1: In general the text is well written, clear and orderly. The topic is an interesting one and falls within the theme of the journal Brain Sciences. The manuscript is scientifically sound and the results can be reproduced. The results of the manuscript are of average novelty, the authors previously made visible part of the results presented in the paper. 

Response 1: We express our sincere gratitude for the positive remarks you have provided regarding our research.

Point 2: Figures/tables/images/diagrams are appropriate. But it is suggested to draw up an experimental design in the form of an image/Scheme, for a better understanding of the experimental procedures.

Response 2:

Based on your recommendation, a study flowchart has been created as depicted in Figure 1. Thank for your help!

Point 3: The conclusions are in agreement with the results of the study.

Response 3:

Thank you for your positive comments

Poin 4: Minor corrections:

RESULTS

Alzheimer's disease (AD) - one line below is written Alzheimer's disease. It is suggested to use the abbreviation after its explanation.

Response 4: We have changed the second “Alzheimer's disease” to its abbreviation (with a blue background). Thank you for your careful erratum.

Reviewer 3 Report

This prospective cohort study aimed to develop and validate a 3-year survival prediction model for patients with cognitive impairment, utilizing demographic data, neuropsychological assessments, and comorbidity information. A total of 616 patients were included, with variables such as age, Mini-Mental State Examination (MMSE) score, Instrumental Activities of Daily Living (IADL) scale score, presence of disinhibition, and Charlson Comorbidity Index being identified as significant predictors of 3-year survival. The model demonstrated strong discrimination and calibration in both training and validation cohorts, with C-indices of 0.80 and 0.85, respectively. This model holds clinical implications by offering accurate survival prediction for patients with cognitive impairment, making it particularly valuable for use in community hospitals and primary care settings without advanced resources. The following comments should be addressed.

Introduction

The authors effectively highlights the importance of early planning for disease management and emphasizes the role of prediction models in estimating life expectancy.

However, there are a few aspects that could be further clarified and expanded upon in the introduction. The authors briefly mentions the limitations of existing prediction models based on cerebrospinal fluid, blood biomarkers, and imaging techniques. While the discussion of these limitations is pertinent, additional elaboration on the specific challenges and drawbacks of these models, as well as their current clinical utility, would enhance the understanding of why a new prediction model is needed.

Furthermore, the authors states that comprehensive neuropsychological assessment data are available for most patients with cognitive impairment and highlights the relevance of scales like MMSE, IADL, and NPI. It would be beneficial to provide more context on the widespread use of these scales, their reliability, and their limitations in predicting prognosis. Additionally, addressing the current gaps in understanding which symptoms from these scales are associated with prognosis, as mentioned, could be elaborated upon to emphasize the specific need for the proposed new prediction model.

Methods

The "Materials and Methods" section provides a comprehensive overview of the study's design and procedures; however, there are several areas that could benefit from further clarification and detail:

-While the inclusion criteria for patients with cognitive decline are described based on MMSE scores and education levels, it would be helpful to explicitly mention the specific MMSE cutoff scores for patients with more than 11 years of education. Additionally, it would enhance clarity to state the rationale behind using these particular MMSE cutoff points.

-Provide more information on the administration and process of the neuropsychological assessment. Explain how the Lawton and Brody IADL scale and the NPI questionnaire were administered, including any specific instructions given to participants.

- Detail the process of calculating the Charlson Comorbidity Index (CCI) based on complications. Specify the nature of these complications and their inclusion criteria in calculating the CCI.

-Offer a brief rationale for choosing a 7:3 ratio to divide the cohort into training and validation datasets. Clarify the specific categorical variables used in the Chi-square test and elaborate on the variables included in the single-factor analysis.

Discussion

-The "Discussion" section provides a comprehensive interpretation of the study's findings and their implications. However, there are several areas that could benefit from further clarification and expansion:

-While the authors discuss the practicality and accessibility of the developed model for primary health care facilities, a more detailed discussion regarding the potential challenges and limitations of applying the model in diverse clinical settings would be valuable. Address potential barriers and considerations that might arise when implementing the model outside of the study's specific memory clinic.

-The authors compare their model with previous predictive models, highlighting the strengths of their approach. However, it would be beneficial to also address any potential limitations or shortcomings of existing models that their approach aimed to overcome. This would provide a more balanced perspective on the contributions of the current study.

-While the discussion touches on the clinical implications of certain findings (e.g., disinhibition's impact on survival), it would be beneficial to delve deeper into the potential implications for patient care and treatment strategies. How might the identification of disinhibition as a prognostic factor influence clinical decision-making and interventions?

-Discuss the importance of external validation to confirm the model's utility in different patient populations and healthcare settings. Additionally, explore potential strategies for addressing the limitations of the study's single-center data and cultural norms when extending the model to broader populations.

-While the limitations of the study are acknowledged, further exploration of how these limitations could affect the generalizability and robustness of the model would enhance the discussion. Additionally, expand on potential avenues for future research, such as refining the model to incorporate factors like diverse etiologies, clinical stages, and treatment modalities.

-Elaborate on how the developed nomogram could potentially be integrated into clinical practice. How might healthcare providers use this tool to guide patient care, especially in scenarios where advanced equipment is not readily available?

Reviewer 4 Report

The authors present a paper aimed at identifying factors influencing the survival of patients with cognitive dysfunction. The vast majority of patients with Alzheimer's dementia took part in the study. Other types of dementia were relatively rare. It should be considered whether the results can be generalized to the entire population, or whether it would not be better to create a model dedicated to Alzheimer's dementia.

At the inclusion stage, the presence of cognitive impairment was identified using only the MMSE scale. Did the patients have other tests performed at this stage to increase the certainty of the diagnosis?

Overall, the study is clearly described and presented. It provides insight into factors contributing to the survival of patients with cognitive impairment.

Round 2

Reviewer 3 Report

I would like to thank the authors for this sufficient revision. Thus, I recommend to accept this paper.